# Exploring Influencing Safety and Health Factors among E-Waste Scavengers in Accra, Ghana

**Opoku Agyemang Addae** [1]**, Haya Fahad Alomirah** [2]**, Heba Faisal Sulaiman Alkhliefi** [2]**, Ravi Rangarajan** [3] **and Haruna Musa Moda** [1,3,]*

1 Department of Health Professions, Manchester Metropolitan University, Manchester M15 6BG, UK; opoku.addae@stu.mmu.ac.uk

2 Department of Natural Science, College of Health Sciences, Shawaik P.O. Box 1983, Kuwait; hf.alomirah@paaet.edu.kw (H.F.A.); hf.alkhliefi@paaet.edu.kw (H.F.S.A.)

3 Department of Environmental Health and Safety, University of Doha for Science and Technology, Doha P.O. Box 24449, Qatar; ravi.rangarajan@udst.edu.qa

* Correspondence: haruna.moda@udst.edu.qa

**Abstract:** The reduced life cycle of electrical goods has contributed to a fast-growing occupational and environmental health issue arising from increased electronic waste (e-waste) in most developing countries. E-waste is processed informally in these countries, and in most cases, it is beyond the reach of official governance and is characterized by a lack of regulation, structure, and any form of license to operate. Assessing the implications of e-waste recycler safety knowledge, awareness, and practice levels is seen as a panacea for developing tailored interventions. We performed a cross-sectional study among 323 e-waste workers located in Agbogbloshie waste dumpsite, Accra, Ghana, to measure their knowledge, awareness, and practice, as well as their perceived safety and behavioral control. A significant negative correlation was found between participants safety knowledge and their safety practices (r (323) = −0.19, p < 0.01), as well as a similar correlation with perceived safety control among the study group (r (323) = −0.27, p < 0.01). In addition, the hygiene rating among the group was adjudged poor as there was no established relationship found with their perceived safety control. To help bridge the gap around e-waste workers safety knowledge, awareness, and practices, it is pertinent for local and international players to take into consideration the shared values and beliefs among the group and work alongside the group in developing a set of policies that will help improve their safety and health.

**Keywords:** informal workers; behavior; knowledge; practices; wellbeing; global south

## 1. Introduction

Over the past decade, Ghana has been a major destination for used electrical and electronic equipment (UEEE) from the global north and is rightly termed the world's electronic graveyard. These e-wastes arrive in the country via Port Tema, located 20 miles east of the Agbogbloshie dump site. Available data reveals that around 53.6 million metric tons (MT) of e-waste were generated globally in 2019, and this trend is increasing at the rate of 2 MT per year [1–3]. Exposure to this e-waste can lead to a number of unwanted health outcomes, like lung-related problems, thyroid imbalance, a decrease in reproductive health, and poor mental and neurodevelopment in the community and site workers [4,5]. The present trend calls for efficient management development that will guarantee the safety and health of e-waste handlers, especially in the global south, the majority of which fall in the global south. In these regions, there is a complete absence of adequate treatment facilities for the management of the products that have reached their end of life and have been transported from the global north [6–8]. The design complexity, along with other factors like a short lifespan, limited repair options, the cost of repairs, and their chemical constituents [9,10], makes it less easy for the electronic products to be recycled. Lack of

recyclability directly leads to new emerging environmental challenges and coupled health problems, which are predominantly shifting away from the source in the global north to the receiving countries, mostly located in the global south. Generally, e-waste exposure sources can be grouped into two sectors, namely informal recycling and formal recycling, and persistent compounds released from either of the two processes reach the environmental media [5]. Most of this e-waste is processed informally, and in most cases, it is beyond the reach of official governance. As a result, this sector is characterized by a lack of regulation, structure, and any form of license to operate [5,11]. It is estimated that in countries like Ghana and other African countries, the formal recycling sector may contribute as little as 0.9% of the overall e-waste recycling [1].

In Ghana, e-waste recycling by the informal sector is mostly carried out by migrants from rural areas, using primitive methods such as open-field burning of products to extract reusable components like copper wire, exposing the e-waste scavengers to direct adverse health effects and pollution of the environment [12–15]. This form of rudimental recycling is not only common in the country but also performed in other countries with shared economic settings, such as Ethiopia [16], Nigeria [9], and South Africa [17], where employment opportunities are becoming much harder to find among the groups involved in the recycling of these products. In general, e-waste contains a mixture of different heavy metals and substances that are considered hazardous to health, and these can easily pollute any environmental medium when they are not processed or recycled sustainably. The common ones are cadmium (Cd), mercury (Hg), lead (Pb), thallium (Tl), chromium (Cr), arsenic (As), and chlorinated compounds that are commonly classed as persistent organic pollutants (POPs) [5,10,17].

Within the informal e-waste recycling sector, heavy metal poisoning can occur through occupational exposure, ingestion of contaminated food, or inhalation of dust, smoke, or fumes in the environment [18,19], leading to an array of health effects that include cancer, respiratory, cardiovascular, and other forms of life-threatening diseases. The potential for such incidences is well documented in Ghana [20,21]. Earlier studies have demonstrated that e-waste scavengers are not fully aware of the health and environmental impacts associated with unsafe processing of e-waste, and this lack of awareness is due to their limited knowledge of e-waste, unsafe recycling, and disposal methods [10,19,22,23]. Several factors have been proposed that offer an explanation for why abled and active individuals resort to e-waste scavenging and recycling activities. Chief among these factors is the limited access to formal jobs and opportunities, especially among the deprived communities mostly located in villages and on the fringes of most urban centers in these developing countries [24,25]. In addition, there exists the notion of perceived safety among these workers, arising out of their judgment of the risks involved and the anticipated level of work safety, which needs to be tested [9,16].

With the rise in environmental and human health challenges associated with informal e-waste management in Ghana and beyond prohibiting the activities of these e-waste scavengers in the informal sector, there is a need for the development of sustainable strategies that could be implemented and carried out by this group of individuals towards the efficient management of the e-waste environmental and health impacts in line with Ghana's government's drive around its SDG goal 3: reduction of non-communicable disease mortality; goal 8: decent work across different genders and age groups; and goal 11: focus around management of air quality and waste management in general. Ghana is a signatory to the Basel Convention and other international treaties [26], with local legislation aimed at the control and management of e-waste and classified waste; however, weaknesses around existing policy enforcement seem to be a major panacea for e-waste activities among its youths that have yet to be effectively managed. To achieve this objective, the paper assessed the implication of e-waste recycler safety knowledge, awareness, and practice levels located at a major dumpsite in Accra, Ghana.

## 2. Material and Method

### 2.1. Study Area Description

The Agbogbloshie e-waste recycling and disposal site is located close to the central business district of the capital city, Accra. All outdoor activities related to informal e-waste recycling like scrap metal collections, dismantling of e-waste parts (using a hammer and chisel), or the recovery of items considered valuable by stripping the material coatings and plastics using available materials such as single-use plastic bags, tires, and dried materials that can serve as a heat source to enable quick melting away of the coating for end product recovery (Figure 1).

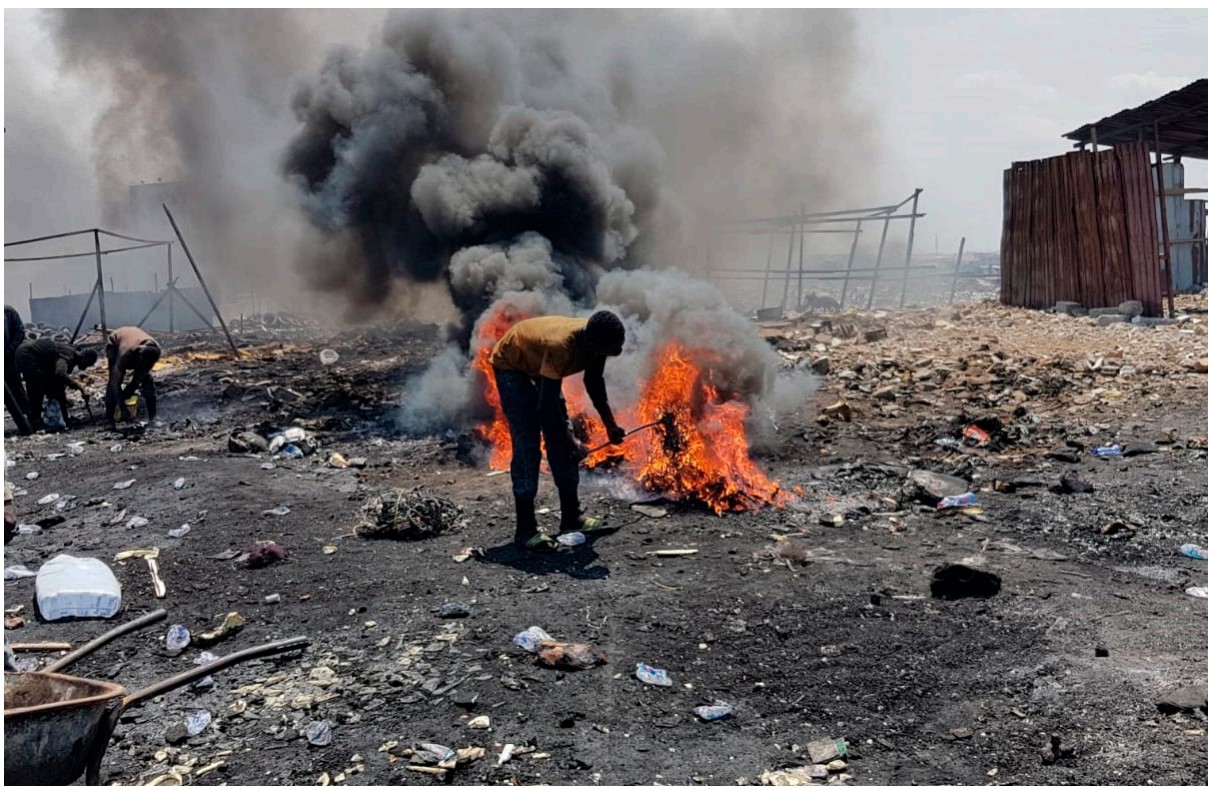

**Figure 1.** Recovery of desirable e-waste materials after burning plastic coatings.

While there is no official figure regarding the number of individuals working in the sector, an earlier study estimated over 30,000 individuals working within the broader e-waste sector [26]. Fisher's formula [27] for estimating single proportions was used to estimate the minimum sample size, and 380 participants were considered an adequate number required for the study. At the end of the data collection period, 351 responses were collated, and after cleaning the data gathered, 323 were used to inform the study outcome. The questionnaire was in English, and as the majority of the respondents were not native English speakers, the primary author was present on site to clarify any further questions regarding the study. Due to variation in the participants work patterns, the researcher administered the questionnaire on site during their rest period, and the approximate time taken for each to respond to the questionnaire was ~30 min.

### 2.2. Data Collection

A cross-sectional study design based on the snow bowl sampling method involving the use of questionnaires and field observation was adopted for the collection of primary data among the target population between April and June of 2021. The faculty of health and education ethics committee at Manchester Metropolitan University granted ethical approval, while participants provided written informed consent before their inclusion

in the study. Prior to administering the questionnaire, a participant information sheet containing the study rationale was provided to each subject at the dumpsite, and further explanation was offered where needed to clarify questions asked regarding the study. Each participant was allowed to decide whether to take part in the study.

Eligibility for the study's inclusion criteria comprised individuals who have spent more than 1 year working as either e-waste collectors, dismantlers of e-waste products, or engaged in the recovery of valuable end products and are above the age of 18. Considering most scavengers were found to reside either within or a short distance away from the dumpsite, a 5-mile radius was set as the boundary for the study's inclusion.

The research survey approach adopted a validated questionnaire consisting of 35 sets of questions to measure their demographics, knowledge, awareness, practice, perceived safety, behavioral control, and hygiene habits. This approach helps to determine participants' knowledge of hazards, their risk awareness, and their judgment regarding the chance of injury or illness associated with the task.

Participants knowledge awareness and perceived safety control were subjectively assessed on a five-point Likert scale with categories of "Strongly Disagree (1)," "Disagree (2)," Neutral (3)," "Agree (4)," or "Strongly Agree (5)", respectively, while their safety practices were measured using the "Yes" and "No" response approaches. Another set of questions did provide a list of potential pollutants to understand the awareness of e-waste handlers towards pollutant associations and the e-waste recycling process. Taking into consideration campaigns held by several stakeholders to raise awareness regarding the public and environmental impacts of improper e-waste importation and recycling processes taking place in Ghana, participants were presented with a list of contaminants to identify those they could associate with the task they were involved with to help further measure their hazards and risk awareness related to the process.

### 2.3. Statistical Analysis

All statistical analysis was undertaken using the statistical package for social sciences (SPSS) 25.0 software (IBM SPSS Statistics for Windows, IBM Corp., Armonk, NY, USA). Descriptive analyses were conducted, and the results are reported as either means, standard deviations, percentages, or frequency tables. Bivariate statistical analysis using Spearman's correlation coefficients was performed to determine the existence of any statistical relationship between two different variables considered in the study. In order to compare the means between participants influencing factors toward the promotion of effective participant safety compliance within the sector, a one-way analysis of variance (ANOVA) was performed. Statistical significance was set at $p < 0.05$.

### 3. Results

A total of 323 e-waste handlers involved in several forms of activities that include door-to-door collection of e-waste (4.6%); buying materials collected by door-to-door scavengers (8.7%); being employed to disassemble sourced/purchased e-waste (10.8%); burning of sourced appliances (43.3%); dismantling and selling of end products (26.8%); and buying of processed (5.8%), respectively, were recruited for the study. Based on the assessed responses obtained, the majority of the participants were male (98.8%), and only 1.2% identified themselves as female. It was also evident that 79.3% of the e-waste handlers were within the age band of 18–23 years. While assessing the participants level of educational attainment, 44.9% were identified as not having any formal education, while 48.9% affirmed having completed their primary education. In addition, 79.6% of respondents identified themselves as having worked for 5 years or less within the informal e-waste recycling sector, with another 17% saying they had worked between 6 and 10 years, and only a fraction having more than 10 years' work experience within the sector. The use of personal protective equipment (PPE) was not actively applied, and 79.3% of the respondents said they did not use any form of PPE while undertaking their task. In addition, a high percentage of

the workers said they do not seek a health check-up (94.7%), and 92.6% said handwashing practice is only performed occasionally before meal breaks (Table 1).

**Table 1.** Participants socio-demographic and occupational characteristics.

| Variables | Frequency | Percentage |
|---|---|---|
| Age in years | | |
| 18–23 | 256 | 79.3 |
| 24–29 | 55 | 17 |
| 30–35 | 10 | 3.1 |
| 41–45 | 2 | 0.6 |
| ≥46 | - | - |
| Gender | | |
| Male | 319 | 99.8 |
| Female | 4 | 1.2 |
| Education | | |
| No formal education | 145 | 44.9 |
| Primary school | 158 | 48.9 |
| Secondary school | 20 | 6.2 |
| Years working in the sector | | |
| 0–5 | 257 | 79.6 |
| 6–10 | 55 | 17 |
| 11–15 | 10 | 3.1 |
| ≥16 | 1 | 0.3 |
| Work pattern | | |
| Full day (>8 hrs) | 323 | 100 |
| Half day (<4 hrs) | - | - |
| Activity type undertaken | | |
| House-to-house collection | 15 | 4.56 |
| House-to-house collection and dismantling | 28 | 8.7 |
| Paid to dismantle on site | 35 | 10.8 |
| Dismantling and sale of parts for processing | 86 | 26.6 |
| Buying processed (burnt) products | 19 | 5.9 |
| Involve the whole processing lifecycle | 140 | 43.3 |
| Living on site | | |
| Yes | 320 | 99.1 |
| No | 3 | 0.9 |
| Distance away from the workstation | | |
| >1-mile radius | 318 | 98.5 |
| 1–5 mile | 4 | 1.2 |
| >5 miles | 1 | 0.3 |
| Use of personal protective equipment | | |
| Yes | 67 | 20.7 |
| No | 256 | 79.3 |
| Health checkup | | |
| Yes | 17 | 5.3 |
| No | 306 | 94.7 |
| Access health services when unwell | | |
| Yes | 311 | 96.3 |
| No | 12 | 3.7 |
| Hand washing before meals | | |
| Always | 24 | 7.4 |
| Sometimes | 299 | 92.6 |
| Never | - | - |

**Table 1.** *Cont.*

| Variables | Frequency | Percentage |
|---|---|---|
| Eat a meal in between work | | |
| Always | 304 | 94.1 |
| Sometimes | 19 | 5.9 |
| Never | - | - |
| I consume water while working | | |
| Always | 323 | 100 |
| Sometimes | 0 | 0 |
| Never | 0 | 0 |
| Smoke while working | | |
| Always | 170 | 52.6 |
| Sometimes | 125 | 38.7 |
| Never | 28 | 8.7 |

To better measure the level of our participants' knowledge of hazards associated with e-waste toxicity and environmental pollution, each participant was presented with a list of possible heavy metals likely to be encountered while processing sourced electronic materials (Table 2). From the list, 56.3% of the respondents identified lead as a potential material associated with the process, followed by mercury (44.3%). On the contrary, none of the participants established a relationship between their activities and possible exposure to palladium, thallium, chromium, arsenic, etc.

**Table 2.** Knowledge assessment of heavy metals associated with e-waste recycling.

| Heavy Metal | Yes (%) | No (%) |
|---|---|---|
| Lead | 56.3 | |
| Mercury | 44.3 | |
| Zinc | 11.1 | |
| Cadmium | 1.9 | |
| Beryllium | 7.1 | |
| Palladium | - | 100 |
| Thallium | - | 100 |
| Chromium | - | 100 |
| Arsenic | - | 100 |
| Nickel | - | 100 |
| Iron | - | 100 |
| Molybdenum | - | 100 |
| Vanadium | - | 100 |

To understand common health symptoms experienced among the participants, a range of possible ill health symptoms were presented, and the participants were asked to identify those they could relate to either during or after handling and processing of e-waste products. From the results, 91% of the respondents reported experiencing a prolonged period of coughing in association with the recycling task. In addition, excessive sweating (83.3%); itchy eyes (80.5%); excess phlegm/mucus (74.9%); shortness of breath (66.9%); dizziness (28.5%); and headache (24.5%) were among the most common forms of ill health symptoms experienced among the participants (Figure 2).

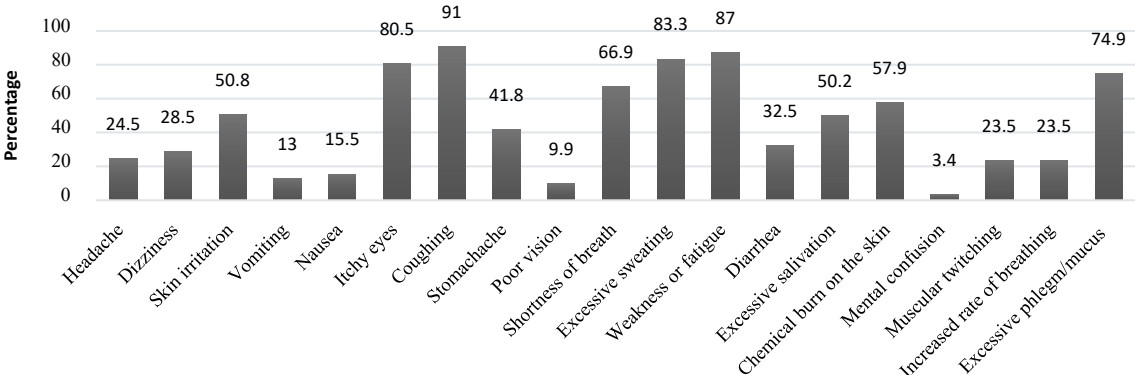

**Figure 2.** Reported health symptoms among participants during and after completion of related tasks.

In order to measure the extent to which scores between the different domains that were assessed are related, an intercorrelation analysis was undertaken, and the results are presented in Table 3. Overall, there were no mutual relationships established between the domains assessed; however, participants safety awareness, safety knowledge, and their safety awareness were found to be positively correlated ($r$ (323) = 0.12, $p < 0.05$). In addition, a positive correlation also exists between the group's safety knowledge and their safety practices ($r$ (323) = 0.155, $p < 0.01$). However, a significant negative correlation was found between participants safety knowledge and their safety practices ($r$ (323) = −0.19, $p < 0.01$), as well as a similar correlation with perceived safety control among the studied group ($r$ (323) = −0.27, $p < 0.01$).

**Table 3.** Inter-correlation matrix for participants safety knowledge, awareness, practices, and perceived safety control.

| | Domains | M | SD | 1 | 2 | 3 | 4 | 5 |
|---|---|---|---|---|---|---|---|---|
| 1 | Safety knowledge | 21.89 | 1.71 | 1 | | | | |
| 2 | Safety awareness | 23.08 | 2.38 | 0.12 * | 1 | | | |
| 3 | Safety practices | 32.43 | 1.36 | −0.19 ** | 0.16 ** | 1 | | |
| 4 | Personal hygiene habits | 11.73 | 0.94 | −0.05 | −0.04 | 0.03 | 1 | |
| 5 | Perceived safety control | 22.88 | 2.36 | −0.27 ** | −0.04 | −0.01 | 0.01 | 1 |

\* $p < 0.05$, \*\* $p < 0.01$, n = 323.

To understand the influence of level of education on participants safety knowledge, awareness, and practices, a one-way analysis of variance (ANOVA) was undertaken. There was a significant effect observed regarding activity types undertaken among the participants across each of the domains measured. Participants age was found to positively influence personal hygiene habits ($F$ (3, 319) = 3.99, $p = 0.008$, $\eta^2 = 0.036$) and their perceived safety control ($F$ (3, 319) = 3.707, $p = 0.012$, $\eta^2 = 0.034$). In addition, education level had a significant effect among the group with regards to safety knowledge, safety awareness, practices, and personal hygiene but not on their perceived safety control ($F$ (2, 320) = 2.185, $p = 0.114$, $\eta^2 = 0.013$). Considering that 98.5% of the participants attest to reside within a mile radius of their work site (Table 1), accommodation location was found to have no significant effect on the participants safety awareness ($F$ (1, 321) = 6.240, $p = 0.013$, $\eta^2 = 0.019$), safety knowledge ($F$ (1, 321) = 0.090, $p = 0.765$, $\eta^2 = 0.000$), safety practices ($F$ (1, 321) = 2.539, $p = 0.112$, $\eta^2 = 0.008$), personal hygiene habits ($F$ (1, 321) = 5.638, $p = 0.018$, $\eta^2 = 0.017$), and perceived safety control ($F$ (1, 321) = 0.162, $p = 0.688$, $\eta^2 = 0.001$) (Table 4).

**Table 4.** One-way ANOVA measured across participants safety knowledge, awareness, practices, hygiene, and safety control.

| | Safety Knowledge | Safety Awareness | Safety Practices | Personal Hygiene Habits | Perceived Safety Control |
|---|---|---|---|---|---|
| Age | $F_{(3, 319)} = 0.751$ <br> $p = 0.522$ <br> $\eta^2 = 0.007$ | $F_{(3, 319)} = 2.307$ <br> $p = 0.077$ <br> $\eta^2 = 0.027$ | $F_{(3, 319)} = 1.279$ <br> $p = 0.282$ <br> $\eta^2 = 0.021$ | $F_{(3, 319)} = 3.996$ <br> $p = 0.008$ * <br> $\eta^2 = 0.036$ | $F_{(3, 319)} = 3.707$ <br> $p = 0.012$ * <br> $\eta^2 = 0.034$ |
| Gender | $F_{(1, 321)} = 1.010$ <br> $p = 0.312$ <br> $\eta^2 = 0.003$ | $F_{(1, 321)} = 0.828$ <br> $p = 0.363$ <br> $\eta^2 = 0.003$ | $F_{(1, 321)} = 2.545$ <br> $p = 0.112$ <br> $\eta^2 = 0.008$ | $F_{(1, 321)} = 19.883$ <br> $p = 0.001$O <br> $\eta^2 = 0.058$ | $F_{(1, 321)} = 0.927$ <br> $p = 0.336$ <br> $\eta^2 = 0.003$ |
| Education | $F_{(2, 320)} = 4.027$ <br> $p = 0.019$ <br> $\eta^2 = 0.025$ | $F_{(2, 320)} = 4.692$ <br> $p = 0.010$ * <br> $\eta^2 = 0.028$ | $F_{(2, 320)} = 5.428$ <br> $p = 0.005$ * <br> $\eta^2 = 0.033$ | $F_{(2, 320)} = 4.589$ <br> $p = 0.011$ * <br> $\eta^2 = 0.028$ | $F_{(2, 320)} = 2.185$ <br> $p = 0.114$ <br> $\eta^2 = 0.013$ |
| Years working in the sector | $F_{(3, 319)} = 0.524$ <br> $p = 0.666$ <br> $\eta^2 = 0.005$ | $F_{(3, 319)} = 2.687$ <br> $p = 0.047$ * <br> $\eta^2 = 0.025$ | $F_{(3, 319)} = 2.806$ <br> $p = 0.040$ * <br> $\eta^2 = 0.026$ | $F_{(3, 319)} = 3.124$ <br> $p = 0.026$ * <br> $\eta^2 = 0.029$ | $F_{(3, 319)} = 4.039$ <br> $p = 0.008$ * <br> $\eta^2 = 0.037$ |
| Activity type undertaken | $F_{(5, 317)} = 12.153$ <br> $p = 0.001$ * <br> $\eta^2 = 0.161$ | $F_{(5, 317)} = 7.315$ <br> $p = 0.001$ * <br> $\eta^2 = 0.103$ | $F_{(5, 317)} = 6.652$ <br> $p = 0.001$ * <br> $\eta^2 = 0.095$ | $F_{(5, 317)} = 3.620$ <br> $p = 0.003$ * <br> $\eta^2 = 0.054$ | $F_{(5, 317)} = 6.415$ <br> $p = 0.001$ * <br> $\eta^2 = 0.092$ |
| Living on site | $F_{(1, 321)} = 6.240$ <br> $p = 0.013$ * <br> $\eta^2 = 0.019$ | $F_{(1, 321)} = 0.090$ <br> $p = 0.765$ <br> $\eta^2 = 0.000$ | $F_{(1, 321)} = 2.539$ <br> $p = 0.112$ <br> $\eta^2 = 0.008$ | $F_{(1, 321)} = 5.638$ <br> $p = 0.018$ * <br> $\eta^2 = 0.017$ | $F_{(1, 321)} = 0.162$ <br> $p = 0.688$ <br> $\eta^2 = 0.001$ |
| Distance away from the workstation | $F_{(2, 320)} = 8.399$ <br> $p = 0.001$ * <br> $\eta^2 = 0.050$ | $F_{(2, 320)} = 3.431$ <br> $p = 0.034$ * <br> $\eta^2 = 0.021$ | $F_{(2, 320)} = 4.302$ <br> $p = 0.014$ * <br> $\eta^2 = 0.026$ | $F_{(2, 320)} = 20.688$ <br> $p = 0.001$ * <br> $\eta^2 = 0.114$ | $F_{(2, 320)} = 0.280$ <br> $p = 0.756$ <br> $\eta^2 = 0.002$ |

Partial Eta Squared ($\eta^2$) = effect size, * $p < 0.05$.

## 4. Discussion

The result of the study shows that e-waste workers at Ghana's waste treatment sites lacked adequate safety knowledge, awareness, and practices to safeguard their health. Using the questionnaire approach and field observation during the data collection, it was evident that the majority of participants lacked the needed support from other stakeholders. In comparison to other studies, there is a strong agreement around the existence of a lack of awareness regarding the health and safety risk and perceived safety control associated with the e-waste processing activities [10,16,28]. Hence, education and years working in the trade are likely influencing factors in regard to potential health risks and knowledge associated with the activities they are willing to engage in during e-waste handling.

Heat exhaustion and general health concerns were noted among the e-waste handlers that took part in the study, of whom 83.3% reported excessive sweating while undertaking the processing task, and 87% of the e-waste workers said they experienced extreme fatigue just after the completion of the work, in addition to other commonly reported health conditions such as cough, excess phlegm, and itchy eyes. These symptoms have earlier been reported among e-waste handlers in other studies too [11,29–31]. While it was not possible based on the present study outcome to conclude that these symptoms exhibited are directly associated with exposure to air pollutants at work, the number of individuals that reported experiencing these states is a demonstration of some form of synergistic relationship between the exposure and other underlining health problems that are likely to present the possibility of non-communicable disease in the future. Building on this, other studies have also demonstrated the impact the workplace could have on individuals and influence both their physical and mental wellbeing [32]. In view of the observation that a high proportion of the worker group engaged in the e-waste recycling activity falls within the young adult window (18–23 years), it is safe to conclude that there exists a direct correlation between a lack of formal employment opportunity and their willingness to engage in informal e-waste recycling as a means of survival despite the poor state of the workplace. In corroboration, recent studies have opined on the existing relationship

between unemployment and poverty as key drivers that lead young adults to engage in informal activities such as e-waste processing, and their financial and economic state is linked to additional stress impacting their mental wellbeing [32–34].

Considering the different types of activities the e-waste handlers on the Agbogbloshie waste dump site are involved in, exposure to hazardous e-waste can occur either through the informal recycling activities undertaken or exposure to e-waste compounds that persist in the environment via inhalation, ingestion, or dermal contact by individuals. Based on the results of the study, e-waste handlers' knowledge of associated e-waste pollutants was found to be relatively low or literally nonexistent. A list of heavy metals associated with e-waste recycling was presented to each participant to identify possible compounds either as components of the material or emitted during the recycling process known to them, and only a sizeable number of the participants were able to associate lead and mercury as compounds associated with hazards related to e-waste recycling. The significant impact of limited knowledge of these hazards is associated with the lack of formal education among the group, as evidenced in previous studies [35,36], which has a great impact on individual related attitudes and practices.

Personal hygiene displayed among the group was found to be an important factor in measuring the group's safety practices. From the result, it was evident that the hygiene rating among the group was adjudged poor as there was no established relationship found with their perceived safety control, and most believe that the work undertaken does not pose any health problem despite 51.8% of participants saying to have experienced skin irritation and 32.5% having diarrhea at some point. Furthermore, the majority consider the use of PPE such as coveralls, respiratory protection, gloves, and boots to be less important as it slows them down when undertaking their job. Health-seeking behavior among the group was low, as the willingness to go for a health check-up was almost absent among the group, and they are only willing to seek medical help when they consider their health critical. This is partly due to mistrust of medical personnel and discriminatory tendencies experienced among the group, as observed in previous studies [35,37]. Hence, there exists a need to expand on the role played by workers health-belief-related interventions to advance and improve their work-life balance.

In addition, based on field study observation, these e-waste handlers are living in an extreme poverty state that is characterized by several factors: income and instability of resources, job insecurity, and the absence of any form of social welfare or amenities, thereby presenting a greater threat to this group's ability to break out of their circle [33]. To achieve desired health, social, and environmental goals, intervention will be required around the provision of better work equipment, infrastructure, and regulations while ensuring these youths have access to basic amenities and related safety training as an avenue to increase their safety awareness, self-esteem, and wellbeing [10,33,38], which can go a long way in reducing the burden of non-communicable diseases associated with air pollution related to e-waste processing. Considering that the majority of the workers fall within the class term "young workers" and are more vulnerable to occupational accident and disease exposure due to the work type undertaken while lacking social protection coverage [39], there is a need for the Ghanaian government, as part of its fundamental principles around occupational health and safety, to consider the integration of workplace safety training at the secondary school level to help strengthen safety behavior among the youth at an early stage of their development. This will help facilitate youths transition from school into decent work [40,41].

While the Agbogbloshie e-waste recycling site has now been cleared as part of the government's urban renewal problem within the area, an alternative work site has been allocated on the outskirts of Accra that offers the opportunity for better interventions to be introduced. With the relocation of the site away from public view, a better approach towards the development and implementation of safety and health policies will help improve working conditions among the e-waste workers in the new workplace.

## 5. Conclusions

While several bodies might hold the view that, with the expansion witnessed in the modern economy, the informal sector, just like e-waste handlers, would gradually see greater improvement around health and safety delivery, the contrary remains the case, especially in the global south. The problem of e-waste generated in the global north has contributed to the environmental and health challenges witnessed within the informal e-waste recycling sector here in Ghana. Taking into consideration the occupational exposure risk and the possibility of incidence occurrence among the e-waste handlers, it is important to acknowledge the effect of the nonexistence of policies and regulations on the e-waste trade, and its impact is likely to be transferred further up and affect the country's goal of meeting its coverage of essential health service interventions around non-communicable disease in the community. Relatedly, within the informal e-waste recycling sector, based on the assessment made here, it was evident that it is essential to bridge the gap around e-waste workers safety knowledge, awareness, and practices. It is pertinent for all actors to take into consideration the shared values and beliefs among the group and work alongside the group in developing a set of policies that will help improve their safety and health. Furthermore, it is of immediate importance to transform the sector into a semi-formal organization to enable the ease of policy and regulation implementation while ensuring educational programs and interventions are established to aid in the reduction of pollution and its associated health risks among the group.

**Author Contributions:** Conceptualization, O.A.A. and H.M.M.; data curation, R.R.; formal analysis, H.F.A. and R.R.; investigation, O.A.A.; methodology, O.A.A. and H.M.M.; project administration, H.M.M.; resources, H.M.M.; software, R.R. and H.M.M.; supervision, H.M.M.; validation, H.F.A., H.F.S.A. and R.R.; visualization, O.A.A., H.F.A., H.F.S.A. and R.R.; writing—original draft, O.A.A.; writing—review and editing, R.R. and H.M.M. All authors have read and agreed to the published version of the manuscript.

**Funding:** This research received no external funding.

**Institutional Review Board Statement:** The study was conducted in accordance with the Declaration of Helsinki, and approved by the Ethics Committee of Department of Health Professions, Manchester Metropolitan University, UK (protocol code 34278, 18 May 2021).

**Informed Consent Statement:** Informed consent was obtained from all subjects involved in the study.

**Data Availability Statement:** The data presented in this study are available on request from the corresponding author. The data are not publicly available to protect participant confidentiality.

**Conflicts of Interest:** The authors declare no conflict of interest.

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
