# Peer review of "Exploring Influencing Safety and Health Factors among E-Waste Scavengers in Accra, Ghana"

_2673-947X, doi:10.3390/hygiene3020017_

Round 1
Reviewer 1 Report
The paper is interesting and significant information can come from that study. However, some lacks should be well explained and the authors should go deeper regarding the method section.
1) Method: the method is not well described, the authors shall go much deeper in that section.
1.1)The authors did not share the questionnaire as an appendix, and it should be done.
1.2) How could/which method the authors used to guarantee that only the target population (from Ghana, above 18 years old, 1 year working as a scavenger, 5 miles, etc) answered the questionnaire? Were the authors (in person) who applied the questionnaire? How much time does it last? How many questions? Was the questionary was applied at the dump site? Before, during, or after the scavenger's work? How was the questionnaire delivered to each person?
1.3) Which language was adopted? English? In Ghana, several languages are spoken. So, if English was adopted, was everybody who answered the questionnaire an English speaker?
2) Results
2.1 ) Regarding table 2 and its paragraph immediately before (142-148)
The authors wrote “…each participant was presented with a list of possible heavy metals…”. And the table shows that the participants only could associate lead and Mercury as heavy metals encountered in e-waste.
In addition, the authors affirm (lines 132-133) “44.9% were identified as not having any formal education while 48.9% affirmed to have completed their primary education”
Therefore, taking into account the worker's education level, I ask the authors: How can the participants know about Vanadium, Thalium, Palladium, etc? Is table two needed? What the table 2 add to the manuscript?
2.2 ) Some results, such as in table 3, “domains” are not described in the method section. The meaning of each domain should be described, even shortly. Why those 5 domains were used? How it was addressed in the questionnaire?
2.3) Table 1 Should be better organized. The variable´s name such as “ age in years”, “gender” etc should not be placed in the same column as the answers option (18-23/24-29/male-female/ etc). Since the sample (quantity/number of participants) is told, the column “frequency” is unnecessary - hold the percentage.
3) Discussion
3.1) Are the symptoms shown in Figure 2 according to the toxicology of heavy metals presented in table 2? The correlation should be presented.
3.2) Which personal protective equipment should be used by the workers?
4) Minor issues
4.1) How the e-waste suck as from Europe (North) can arrive in Gana?
The authors affirm that e-waste is shifting away ( North to south). It should be better explained.
4.2) Line 57 – the element Lead should be Pb (not Ld)
4.3) Lines 106-122- those paragraphs Should be formatted
4.4) The skipped line between the paragraph and table should be adjusted
4.5) The reference section should be formatted
Author Response
Dear Reviewer,
Many thanks for your valuable comments. Find attached our response to each observation raised
Best regards

Reviewer 2 Report
General
The researchers ought to be commended for choosing an important research topic. It is noteworthy that the case (Ghana) selected is one that has the potential to yield valuable insight on the research topic. Generally, the presentation is good.
I have offered a few comments to help the researchers to strengthen the paper.
Introduction
The researchers need to set the hook in this section of the manuscript by paying attention to the following:
· Briefly mention why Ghana was chosen as a case study. You’ll later have to expand on this in the methods section (see my comments on the methods section). I know that Ghana is one of the best cases for exploring the research topic, but you can’t assume that your readers already know.
· Problematize the issue under investigation more deeply because this is how you will get the attention of researchers and policymakers. As it is now, you attempt to do so by mentioning some of the problems associated with e-waste scavenging. If there are any sources that provide statistics to stress the seriousness of these problems, please cite them. What does Ghana, Global South counties, and the world sand to lose if the current situation continues unabated.
· Overall, briefly mention what Ghana and other countries, especially those from which used electrical goods are imported, stand to benefit from your research
Your paper deals with the issue of safety, so I suggest it would be good to engage with the safety literature more fully. For instance, “perceived safety” is a theoretical concept in the health and safety literature, so at least I expect to see some literature review on this as well as on other concepts such as perceived safety behavioral control, safety knowledge, etc. Try to unpack the meanings of some of these terms and how they apply to your research topic. This will provide a proper theoretical context within which your readers can situate your findings. It will also enable your research to be assessed for its theoretical contributions.
The use of the term LMIC or Global North/South must be consistent throughout the paper. I suggest the latter would be more appropriate in this case.
Materials and method
There is every indication that this research is a case study. It would therefore be useful to provide something brief on the case study method and why it was selected.
While some information is provided under the heading “Study Area Description”, it will further strengthen your paper if you expand on the case description. This is good for bringing the research context alive. Give some good background about the Agbogbloshie area and the issue of e-waste. Furthermore, provide some justification as to why it was selected as a case (what makes it really unique and how did selecting this case enhance your ability to undertake your research?)
The specific numerical weights of the Likert items must be indicated.
How did you arrive at the sample size?
Results
This section is adequately presented.
Discussion
The results indicate the target population is predominantly young. It is good that this has been pointed out and possible the underlying reason explained.
What are the implications of this finding for research and practice? Do interventions need to be youth-led? Do interventions need to be designed to make them more appealing to youth? I’m asking this because in the domain of health and safety, young people are known not to patronize interventions.
You also mention lack of knowledge as a cause of the problems you identified. Could this mean that interventions should focus on improving literacy?
Conclusion
Considering that the majority of the world’s youth live in the Global South, I believe that your research has global relevance and this should be stressed in your conclusion.
Currently, young people’s health (both mental and physical) is key on the global health agenda. This is particularly true in the case of those in the GS. Emphasize how your research adds to policy and research on this group and call for more future research. Perhaps you can even mention the importance of your research in connection with SDGs 3, 8&11.
The references below might be useful:
International Labour Organization [ILO] 2018. Improving the safety and health of young workers. Geneva: International Labour Office.
International Labour Organization [ILO] 2017. Global Employment Trends for Youth 2017: Paths to a better working future. Geneva: International Labour Office.
International Programme on the Elimination of Child Labour [IPEC], 2011. Children in hazardous work: What we know, what we need to do. Geneva: International Labour Office.
Congratulations on your good work. Best wishes.
Please undertake thorough proofreading and improve the grammar and sentence structure.
Pay attention to how punctuations are used in certain paragraphs. A few of them were quite hard to understand without repeated reading.
Author Response

(The authors gave the same response as above.)

Reviewer 3 Report
Thank you for the opportunity to review this interesting paper.
In the section Introduction, the authors should clearly explain the research problem and the objectives of the study. The authors need to articulate the ‘missing puzzle piece’ their research aims to cover more clearly and specifically. I usually recommend the logical flow for the introduction to be: 1) what is the problem and why is it important, 2) what we know, 3) what we don’t know, and finally, 4) what are we doing about it.
Discussion and conclusion: Describe in more detail how a new knowledge could be used in practice. I recommend the authors to articulate clearly what the contributions of the paper are to:
1) Theory – the body of conceptual knowledge
2) Practice – to managers / employees / policy makers
Author Response

(The authors gave the same response as above.)

Round 2
Reviewer 1 Report
This reviewer is pleased. The authors proceeded according to the addressed comments.
Author Response
Many thanks for the observation
Comments have now been updated
Reviewer 2 Report
The authors have adequately responded to all my comments. The quality of the paper is significantly improved now and is good for publication.
A few minor errors need to be corrected. I believe they can be resolved during the final proofreading stage before publication.
Author Response
We are graetful for the comments raised.